# Regulation of the Receptor Tyrosine Kinase AXL in Response to Therapy and Its Role in Therapy Resistance in Glioblastoma

**DOI:** 10.3390/ijms23020982

**Published:** 2022-01-17

**Authors:** Lea Scherschinski, Markus Prem, Irina Kremenetskaia, Ingeborg Tinhofer, Peter Vajkoczy, Anna-Gila Karbe, Julia Sophie Onken

**Affiliations:** 1Department of Neurosurgery, Charité-Universitätsmedizin Berlin, Corporate Member of Freie Universität Berlin, Humboldt-Universität zu Berlin, and Berlin Institute of Health, 10117 Berlin, Germany; Lea.Scherschinski@charite.de (L.S.); Markus.Prem@uniklinikum-dresden.de (M.P.); irina.kremenetskaia@charite.de (I.K.); peter.vajkoczy@charite.de (P.V.); anna-gila.karbe@charite.de (A.-G.K.); 2Department of Neurosurgery, Technische Universität Dresden, 01069 Dresden, Germany; 3Department of Radiooncology and Radiotherapy, Charité-Universitätsmedizin Berlin, Corporate Member of Freie Universität Berlin, Humboldt-Universität zu Berlin, and Berlin Institute of Health, 10117 Berlin, Germany; ingeborg.tinhofer@charite.de; 4German Cancer Consortium (Deutsches Konsortium für Translationale Krebsforschung–DKTK), Partner Site Berlin, 10115 Berlin, Germany

**Keywords:** RTK-AXL, tyrosine kinase inhibitor (TKI), R428, glioblastoma multiforme, radiation, temozolomide, post-translational receptor modification

## Abstract

The receptor tyrosine kinase AXL (RTK-AXL) is implicated in therapy resistance and tumor progression in glioblastoma multiforme (GBM). Here, we investigated therapy-induced receptor modifications and how endogenous RTK-AXL expression and RTK-AXL inhibition contribute to therapy resistance in GBM. GBM cell lines U118MG and SF126 were exposed to temozolomide (TMZ) and radiation (RTX). Receptor modifications in response to therapy were investigated on protein and mRNA levels. TMZ-resistant and RTK-AXL overexpressing cell lines were exposed to increasing doses of TMZ and RTX, with and without RTK-AXL tyrosine kinase inhibitor (TKI). Colorimetric microtiter (MTT) assay and colony formation assay (CFA) were used to assess cell viability. Results showed that the RTK-AXL shedding product, C-terminal AXL (CT-AXL), rises in response to repeated TMZ doses and under hypoxia, acts as a surrogate marker for radio-resistance. Endogenous RTX-AXL overexpression leads to therapy resistance, whereas combination therapy of TZM and RTX with TKI R428 significantly increases therapeutic effects. This data proves the role of RTK-AXL in acquired and intrinsic therapy resistance. By demonstrating that therapy resistance may be overcome by combining AXL TKI with standard treatments, we have provided a rationale for future study designs investigating AXL TKIs in GBM.

## 1. Introduction

Glioblastoma multiforme (GBM) is the most common and most aggressive brain tumor in adults. It remains an incurable disease with a markedly diminished life expectancy despite multimodal therapy comprising surgical resection, chemotherapy, and radiation (RTX) [1,2]. Recent efforts in the treatment of GBMs have focused on individualized treatment protocols aiming to target certain mutations (e.g., Epidermal Growth Factor Receptor (EGFR) mutation, Neurotrophic Tyrosine Receptor Kinase (NTRK) fusion or multiple receptors using multi-kinase inhibitors e.g., Regorafenib (Stivarga, Bayer, Leverkusen, Germany) and Sunitinib (Sutent, Pfizer, New York, NY, USA), with limited success [3,4,5,6]. Molecular mechanisms responsible for treatment failure and therapy resistance are the focus of current investigations [7]. The receptor tyrosine kinase AXL (RTK-AXL) has been identified as a mediator for tumor progression and therapy resistance in various cancer types, including squamous cell tumors, small cell lung cancer, and breast cancer [8,9,10,11,12,13].

RTK-AXL is a transmembrane receptor with a C-terminal intracellular domain and an N-terminal extracellular domain. RTK-AXL belongs to the TAM family of kinases and has a specific ligand, which is the growth arrest-specific protein 6 (GAS6) [12,14]. Binding of Gas6 to the extracellular domain of the receptor leads to (hetero-) dimerization of the receptor/Gas6 complex and consecutive phosphorylation of the intracellular domain. RTK-AXL phosphorylation (P-AXL) results in the activation of Phosphoinositide 3-kinases (PI3K) and its downstream target, serine/threonine protein kinase B (Akt) [15]. The Gas6/AXL/PI3K/Akt pathway protects cells from apoptosis via multiple mechanisms. In particular, Akt activates ribosomal protein S6 kinase (S6K) of the mechanistic target of the rapamycin (mTOR) pathway and phosphorylates BCL2-associated agonist of cell death (Bad), which is a pro-apoptotic protein [7,14].

In GBM, RTK-AXL is widely expressed and activated in primary and recurrent diseases [16]. The activated receptor P-AXL leads to a significant increase in tumor proliferation, tumor cell migration, and angiogenesis in in vitro and in vivo GBM models [17,18]. Furthermore, it has been shown that overexpression of RTK-AXL is associated with poor prognosis in GBM [16]. The oncogenic effects may be reversed by targeted inhibition with tyrosine kinase inhibitors (TKI), as has been previously described by our group and others [16,19,20]. To date, it remains largely unknown whether the overexpression of RTK-AXL occurs in response to therapy or RTK-AXL itself plays a role in therapy resistance in GBM. Consequently, the optimal timing for the use of TKIs in GBM remains to be clarified.

Fortunately, there is now a variety of AXL TKIs available whose effectiveness can be studied in detail in vitro and in vivo [21,22,23]. The most specific RTK-AXL inhibitor is R428 (Bemcentinib, also known as BGB324, BerGenBio, Bergen, Norway). Furthermore, R427 was the first TKI to enter clinical trials in 2014 (NCT02424617, NCT02922777 etc.) and is being used in an early phase I clinical trial in recurrent GBM (NCT03965494) [21,24].

To evaluate the role of RTK-AXL in acquired and intrinsic therapy resistance and to provide a robust rationale for future study designs of clinical trials investigating TKIs in GBM, we aim to: (i) characterize the regulation of RTK-AXL in response to RTX and temozolomide (TMZ) treatment, (ii) investigate whether endogenous RTK-AXL expression is related to therapy response; and (iii) evaluate if the co-administration of the AXL-specific TKI R428 to standard GBM therapy, consisting of TMZ and RTX, augments treatment effects.

## 2. Results

### 2.1. RTK-AXL Undergoes Posttranlational Receptor Modification in Response to Therapy

First, we investigated whether the treatment with TMZ modifies the RKT-AXL expression in glioma cells. SF126 and U118MG cell lines were exposed to different TMZ concentrations, and the corresponding AXL mRNA levels were determined. Only minor changes (less than 1.5-fold) of AXL mRNA levels were observed in both cell lines following exposure to therapeutic doses of TMZ (10 µM) after 24, 48 and 72 h, and was most pronounced at 48 h (Ordinary one-way ANOVA *p* = not significant, Figure 1A,B). Consequently, we analyzed RTK-AXL expression on protein levels with Western blot analysis 48 h after TMZ treatment. Full length RTK-AXL (FL-AXL) and C-terminal AXL (CT-AXL) protein expression increased slightly after single dose TMZ in both cell lines SF126 and U118MG (Figure 1C). Quantification of protein expression is shown in Figure 1D.

To simulate the clinical situation of prolonged TMZ exposure, we now exposed cell lines to increasing doses of TMZ on five consecutive days. Interestingly, FL-AXL decreased and CT-AXL increased in a dose-dependent manner (Figure 1E). Quantification of the protein levels are displayed in Figure 1F (Ordinary one-way ANOVA, **** *p* < 0.0001). This dose-dependent, opposed regulation of FL-AXL and CT-AXL could be interpreted as a post-translational receptor modification in response to TMZ treatment. To prove this observation, we aimed to measure the N-terminal RTK-AXL domain, also known as soluble AXL (sAXL), in the supernatant of the TMZ-exposed cells. In accordance with the increasing CT-AXL expression levels, we detected an increased activity of the tumor necrosis factor-alpha converting enzyme (TACE/ADAM17), a member of the disintegrin-metalloprotease (ADAM) family that is involved in proteolytic shedding of the RTK-AXL releasing sAXL following TMZ treatment [25]. This increase in TACE activity was observed in both U118MG and SF126 cell lines over time, indicating the involvement of TACE in post-translational RTK-AXL modification in response to TMZ (Figure 1G,H).

To determine RTK-AXL regulation in response to RTX, the cell lines SF126 and U118MG were exposed to increasing dosages of RTX (0, 2 and 6 Gray (Gy)). On mRNA level, no significant regulation of AXL mRNA was observed with the most pronounced however not significant changes 24 h after RTX (Figure 2A,B). Corresponding to that, we evaluated FL-AXL and CT-AXL protein expression 24 h following RTX and did not observe any changes (Figure 2C). Quantification of the protein levels are displayed in Figure 2D. Accordingly, the activity of TACE remained unchanged between irradiated cells and control cells following RTX (Figure 2E,F).

To simulate a radio-resistant microenvironment, we cultivated the cells under hypoxic conditions resulting in an increased expression of hypoxia induced factor 1 (HIF1 alpha), a known trigger of radio-resistance [26,27,28]. We observed a significant increase of CT-AXL in a time-dependent manner corresponding to the increase of HIF1 alpha protein expression (Figure 2G, quantification of protein expression Figure 2H) while with FL-AXL expression remained unchanged. Altogether, the above presented data indicate that repeated, increasing TMZ doses (on five consecutive days) and exposure of cells to a radio-resistant microenvironment led to a substantial post-translational RTK-AXL modification, whereas single fraction RTX and single dose of TZM only had minor effects on AXL mRNA levels (post-transcriptional modifications), and RTK-AXL protein expression.

To assess RTK-AXL phosphorylation, downstream pathway activation and activation of alternative RTKs in response to TMZ and RTX, Western blot with activated Akt (pAkt) and STAT (pSTAT) (signal transducer and activator of transcript proteins) and Protein phospho-kinase array was applied. No significant changes of RTK-AXL down-stream pathway activation or phosphorylation of alternative RTKs were observed in response to the therapy (Appendix A).

### 2.2. AXL TKI R428 Combined with Standard Therapy Increases Efficacy of TMZ and Radiation

Irrespective of RTK-AXL modifications in response to therapy and hypoxia, we aimed to determine whether simultaneous RTK-AXL inhibition with TKI R428 and TMZ/RTX could increase therapy response and if TKI R428 could aid in overcoming therapy resistance. Therefore, we established two TMZ-resistant cell lines (SF126-TR and U118MG-TR) in addition to the TZM naïve cell lines (SF126 and U118MG), to study the efficacy of the combination treatment. To create the TR cell line, SF126 and U118MG cells were treated with gradually increasing concentrations of TMZ (10–150 µM) for 10 to 12 weeks. Resistance to TMZ was based on a stable colony formation capacity in SF126-TR and U118-TR cells under TMZ treatment, compared with a significant decrease of viability in naïve SF126 and U118MG glioma cells (Figure 3A,B; paired *t*-test SF126 vs. SF126-TR: ** *p* = 0.006; paired *t*-test U118MG vs. U118MG-TR: ** *p* = 0.025, Figure 3C,D displays respective images of the CFA). Next, we determined 0.5 µM of R428 as the dose at which we could not detect RTK-AXL receptor degradation/shedding, a known mechanism in the development of resistance to TKIs (weak expression of CT-AXL under 0.5 µM R428, Appendix A). Subsequently, we exposed U118MG and SF126 to a subtherapeutic dose of TMZ (5 µM) and 0.5 µM R428. As shown in Figure 3E,F, the administration of either TKI R428 or TMZ had no significant effect on cell viability, whereas the combination of TMZ and R428 reduced survival fractions in CFA in both cell lines significantly (ordinary one-way ANOVA, SF126: *p* = ns, U118MG ** *p* = 0.0069). To study the efficacy of TKI R428 in the established TR-cells, we treated SF126-TR and U118-TR cells with R428 and TMZ and performed a CFA. Interestingly, the CFA revealed a significant reduction in the colony formation capacity in both SF126-TR and U118-TR cell lines with R428 given alone (Student’s *t*-test comparing SF126 with SF126 + R428: ** *p* = 0.0015 and U118MG with U188MG + R428: ** *p* = 0.0028). The combination of TMZ and R428 further increased this effect (Figure 4E,F; ordinary one-way ANOVA, SF126-TR: **** *p* < 0.0001, U118MG-TR *** *p* = 0.0002).

We set out to test the radio-sensitizing capacity of R428 by exposing SF126 and U118MG cell lines to RTX and R428 and quantified cell survival using CFA. The addition of R428 to RTX treatment resulted in a highly significant decrease in colony formation capacity in the SF126 cell line (Figure 4A, ordinary one-way ANOVA, SF126: **** *p* < 0.0001). A similar trend was observed for the U118MG cell line; however, the results were not statistically significant (Figure 4B, ordinary one-way ANOVA, U118MG: *p* = ns). These results indicate that the application of R428 alone in TR cells seems to be effective, whereas the combination of R428 with TZM or RTX leads to an improved therapeutic response even in TMZ resistant cell lines. This in turn supports the hypothesis that RTK-AXL inhibition might increase sensitivity towards TMZ and RTX and has the capacity to resolve mechanisms of therapy resistance.

### 2.3. Endogenous RTK-AXL Expression Is Related to Radiation Sensitivity

Lastly, we aimed to test the hypothesis whether endogenous RTK-AXL expression levels are related to treatment response. Therefore, the following modified cell lines were exposed to TMZ and RTX: SF126-DN: overexpressing the RTK-AXL lacking the intracellular domain (AXL-DN) and SF126-WT: overexpressing the functional RTK-AXL (Appendix A). SF126, SF126-DN and SF126-WT cells were exposed to increasing doses of TMZ. The survival was determined with CFA. As shown in Figure 4C, TMZ treatment did not result in survival differences depending on the endogenous (functional) RTK-AXL expression of the respective cell lines. Next, we exposed cell lines to RTX and measured the survival fraction with CFA. When exposing these cells to 2–6 Gy RTX, CFA revealed that SF126-WT were significantly less sensitive to RTX compared with the control cell line SF126 and SF126-DN cells (Figure 4D, ordinary one-way ANOVA, SF126WT at 2 and 6 Gy: **** *p* < 0.0001). With this data we show that elevated endogenous expression of the functional RTK-AXL is associated with less radiosensitivity, whereas sensitivity to TMZ remains unaffected.

## 3. Discussion

With this study we showed that RTK-AXL undergoes post-translational degradation in response to repeated TZM exposure and under hypoxia, representative for a radio-resistant microenvironment, whereas these effects were not observed following a single fraction of RTX or single dose of TMZ. Further, we showed that high endogenous levels of FL-AXL reduced therapy response to RTX. However, therapy resistance may be overcome by combining AXL TKI R428 to standard treatment, even in TMZ-resistant GBM cell lines.

Post-translational receptor modifications are considered a relevant mechanism of therapeutic resistance to TKIs [25,29]. Additionally, receptor shedding may be induced by standard therapy such as RTX [29]. Our observation that shedding product CT-AXL is increasing under hypoxia, a known contributor to radio-resistance, and in response to repeated doses of TMZ, supports the idea that RTK-AXL receptor shedding plays a role in the development of acquired therapy resistance [10,25,30,31]. From the literature it has been reported that shedding products are capable to regulate transcription and activate feedback loop mechanisms [29]. Interestingly we observed neither RTK-AXL receptor phosphorylation downstream signaling activation nor deactivation, as has been reported for the notch signaling pathway or MerTK [32,33]. Therefore, we hypothesize that RTK-AXL shedding products might be involved in the development of therapy resistance in GBM by an alternative pathway activation or induction of transcriptional effects, as it has been shown by McDaniel et al. and Hong et al. for head and neck squamous cell carcinoma [34,35]. They demonstrated that RTK-AXL has the ability to prevent DNA-damage-induced apoptosis by blocking nuclear translocation of c-ABL in the context of cisplatin and RTX resistance. Based on this data, it seems highly relevant to further explore the role of RTK-AXL shedding products in the context of acquired therapy resistance in GBM in an experimental setting by applying technologies such as spatial transcriptomics and proteomics. Even though we could not show significant post-transcriptional modification of AXL mRNA in our study, this mechanism must be further explored since there is evidence for negative feedback regulation of RTK-AXL by mircoRNAs in response to chemotherapy in lung cancer cells [36].

In the clinical context, the question arises as to the optimal timing for the use of TKIs and appropriate combinational therapies. Our data shows that TKI R428 is acting as a therapy sensitizer to TMZ. The same has been observed in cell lines with acquired resistance to TMZ. When combining TKI R428 and RTX, we observed a significant improved therapy response of the SF126 cell line. These findings indicate that TKI R428 increases sensitivity towards TMZ and RTX and has the capacity to resolve mechanisms of therapy resistance. Since these results could not be reproduced for the combination of RTX and TKI R428 in the U118MG cell line, there might be a rationale for individual testing of patient-derived cell lines in future GBM studies to generate substantial rationales for applying AXL TKIs as single agents or to standard therapy consisting of fractionated stereotactic RTX and TMZ.

Regarding patient stratification for the use of AXL TKIs in clinical trials, we addressed the role of endogenous RTK-AXL expression in GBM. Although we could not establish a correlation between high RTK-AXL expression and the TMZ treatment response, there seems to be a correlation between endogenous AXL expression and radiosensitivity, which corresponds well with previously published studies [37]. We, along with others, have observed that in human GBM patients increased RTK-AXL activation and expression is associated with a shorter overall survival [16,19]. Based on this data, endogenous RTK-AXL expression seems to be correlated with intrinsic therapy resistance especially in the context of RTX. Therefore, consideration of the endogenous RTK-AXL expression of individual GBM patients is highly recommended for the design of clinical trials and patient stratification.

In summary, our data show that RTK-AXL appears to play a critical role in both intrinsic and acquired resistance to standard GBM therapy. Monitoring RTK-AXL expression as well as shedding products as indicators of therapy resistance and tumor progression could be a promising approach for individualized therapy management as it is currently being studied for malignant melanoma and hepatocellular carcinoma patients [22,31].

## 4. Material and Methods

### 4.1. Cell Culture

Human high-grade glioma cells U118MG were obtained from American Type Culture Collection (ATCC). Human high-grade glioma cells SF126 were obtained from the JCRB Cell Bank. Both were primary tumor cell cultures derived from surgical specimens of human GBM, WHO Grade IV. Cell authentication was carried out with LGC Standards Cell line Authentication service in June 2014 with 16 loci service of short tandem repeat profile. Further, we used a truncated form of human RTK-AXL lacking the intracellular RTK-bearing domain into SF126 cells (AXL-DN). SF126 naïve cell line served as control. Western blotting confirmed expression of the wild-type and truncated receptor in AXL-DN cell clones (Appendix A).

Tumor cells were maintained as monolayer cultured in tumor growth medium at 37 °C, 5% CO_2_, 95% humidity in a tissue culture incubator. Growth medium encompassed Dulbecco’s modified Eagle’s medium (DMEM, Invitrogen) supplemented with 10% fetal calf serum (FCS) and 1% antibiotics (penicillin/streptomycin). Each experiment was conducted under the same conditions and repeated in three independent runs. Cell lines were cultivated to 80% confluence before starting an experiment. Therefore, cells were cultivated with 10–15 mL of cell culture medium in a T75 flask or with 5 mL of cell culture medium in a T25 flask. Cells were seeded at a density of 7 × 10^5^ cells per mL medium. The cell culture medium was replaced 24 h after thawing the cells and was then replaced twice a week. Cell counts and cell viability were assessed with CASY^®^ Cell Counter TT (OLS). For hypoxia experiments, cells were cultured under 1% O_2_ for 3, 6, 12, and 24 h accordingly to reports from the literature [38,39].

### 4.2. Colony Formation Assay (CFA)

A known number of cells (250–1000) were plated into a 12-well plate and treated with either TMZ, RTX or combined treatments starting after one day in culture. After a growth phase of 12 to 14 days cells were fixed with 70% ethanol and stained with Giemsa. Surviving cell colonies were counted visually and microscopically. A colony was considered >50 cells. Survival fractions were graphed in relation to vehicle control.

### 4.3. Quantitative Real-Time PCR

Following isolation of total mRNA using the PureLink™ RNA Mini Kit (Invitrogen™, Carlsbad, CA, USA) according to the manufacturer’s instructions, the concentration and purity of extracted mRNA was measured with Tecan Infinite Pro 200 microplate reader (Tecan, Zürich, Switzerland). Reverse transcription was carried out using QuantiTect^®^ Reverse Transcription Kit (Qiagen, Hilden, Germany). Quantitative analysis was performed using QuantiTect^®^ SYBR Green gene expression master mix (Takara Bio Inc., Kusatsu, Japan) and QuantStudio™ 6 Flex Real-Time PCR System (Applied Biosystems, Foster City, CA, USA). Each reaction compound was composed of 5 µL SYBR Green master mix, 4 µL cDNA (10 ng/mL), 0.2 µL ROX lL, 0.2 µL of forward and reverse primers, respectively and 0.4 µL H_2_O yielding a total volume of 10 µL. Expression levels of the housekeeping gene 18S were used to normalize RTK-AXL expression levels and RQ values were graphed in relation to vehicle control. We consider a RQ significant when there is a minimum of two-fold change. The relative mRNA levels were calculated based on the cycling threshold (C_t_) values. Each experiment was repeated in biological triplicates.

Primer design was done with the NCBI primer design tool (https://www.ncbi.nlm.nih.gov/tools/primer-blast/ accessed on 20 January 2017) and according to Roux et al. [40] The following criteria were taken into account: after defining the accession of the mRNA template, the position ranges for forward and reverse primers were entered so that primers were located on specific sites of the target genes. PCR product size and melting temperature (Tm) were determined. Further primers used to transcribe RNA to cDNA needed to span an exon-exon junction in mRNA. Primer length was chosen in between 17–28 bases. The tool takes account of the base composition, which should consist of 50–60% G + C and includes 3’end of a G or C or CG or GC. Tm was calculated between 55–80 °C. Further it was checked that the 3’ends of the primers were not complementary. Runs of three or more Cs and Gs at the 3’end of primers were avoided, as were G- and C-rich sequences. Any additional homology was excluded by blasting primer sequences through the NCBI gene pool database. Following primer sequences have been used from Table 1 (TIB MOLBIOL, Berlin, Germany):

### 4.4. Western Blot

After lysis in Radioimmunoprecipitation assay (RIPA) buffer, samples were frozen and stored in −80 °C. Protein concentrations of the samples were measured using the BCA Protein Assay as instructed by the provider (Thermo Scientific™, Pierce™ BCA Protein Assay Kit). Experiments were run with 20 µgr protein. Western blots were repeated with protein samples from 2–3 independent experiments. Protein samples were boiled for 5 min with Laemmli sample buffer (Bio-Rad Laboratories) and loaded into pre-poured Tris-HCl-glycine SDS-PAGE gels (stacking gel 4%, resolving gel 6%). Gels run at 150 V for 1.5 h following transfer to a polyvinylidene difluoride membrane (PVDF, Bio-Rad Laboratories) at 400 mA constant current for 2 h. Blots were blocked with 5% BSA in 1xTBST, primary and secondary antibodies were dissolved in TBST. Following primary antibodies were used. R&D Systems: human phopho-AXL (Y779) mAb (Clone 713610), Cell Signaling: anti-AXL C89E7 rabbit mAb, abcam: anti-AXL (EPR21107) ab215205 mAb, phospho-MET (Tyr1234/1235) rabbit mAb (D26) XP^®^, HIF-1α Antibody (#3716). Santa Cruz: Gas6 antibody (C-20). Sigma Aldrich: mouse monoclonal β-actin antibody (clone 1A4). Immune complexes were visualized using a second HRP-conjugated anti-rabbit or anti-mouse antibody (Pierce Biotechnology). Pierce™ ECL Western Blotting Substrate (Thermo Fisher Scientific, Waltham, MA, USA) was used to develop blots and ImageJ Software was used for densitometry analysis. Reported values were first normalized to the loading control and then multiplied by a constant to reach the lowest whole integral. Each Western blot was carried out with a negative control consistent of sample diluent or beads incubated with antibody and sample diluent only. Western blot quantification was carried out using ImageJ software (LOCI, University of Wisconsin, Madison, WI, USA) according to the user guide IJ1.46r.

### 4.5. TACE Assay

TACE assay (R&D Systems, Minneapolis, MN, USA, Catalog No. ES003), was carried out as previously described according to manufactures instructions. Cells were lysed in lysis buffer as previously described and total protein content was measured using Pierce™ BCA Protein Assay Kit (Thermo Fisher Scientific, Waltham, MA, USA) [18]. 50 ug protein lysates were incubated with a fluorogenic ADAM10/17 substrate peptide (TACE substrate lll, 10 µM, R&D Systems) and fluorescence was measured with a microplate reader (Tecan, Infinite Pro 200) at 320 excitation and 405 emission every 10 min for 4 h until a plateau was reached. Purified mouse ADAM 17 (R&D Systems, Catalog No. 2978-AD) served as the positive control, and the absence of cell lysate served as the negative control.

### 4.6. Protein Phospho-Kinase Array

The Proteome Profiler™ Array Human Phospho-RTK Array Kit (R&D Systems, Minneapolis, MN, USA, Catalog No. ARY001B) was used according to the manufacturer’s instructions. Protein was harvested in lysis buffer and 300 ug total protein was loaded onto specialized membranes coated with multiple human phospho-kinase antibodies. Detection was performed with GeneSnap program shooting for 20 min with 1 min running time between captures.

### 4.7. Temozolomide (TMZ) Treatment in Cell Culture

TZM for use in cell culture was acquired commercially from Sigma Aldrich (St. Louis, MO, USA). To establish TMZ-resistant glioma cell lines, SF126 and U118MG cells were exposed to gradually increasing concentrations of TMZ (5–150 µM) in culture media for 10 to 12 weeks. TMZ was prepared freshly every day and added as 1% DMSO composite of total volume to fresh tumor growth medium. Vehicle control cells were fed daily with fresh tumor growth medium containing 1% DMSO. For two weeks cells were exposed to low-dose concentrations of 5 µM TMZ. Then, concentrations were increased prior to every other passage when cells had reached 85% confluence. Cells were passaged in half to ensure sufficient survival. Later during high-dose treatment (>80 µM) cells were given multiple passage cycles before rising in treatment concentration. The generated TMZ-resistant cell lines were named SF126-TR and U118MG-TR. Cells were assessed regularly for sufficient viability using colorimetric microtiter (MTT) assay and microscopy (Appendix A).

### 4.8. Radiation Protocol for Cell Culture

Human high-grade glioma cells SF126 and U118 were plated at a defined density measured with the CASY^®^ Cell Counter TT (OLS) in 12-well plates for CFA or 25 cell culture flasks for Western Blot and PCR experiments and received different RTX doses (1, 2, 4 or 6 Gy) after 24 h in culture. Applied RTX doses were determined according to other studies investigating RTX effects in glioma cells [41]. Cell damage in SF126 glioma cells had been shown with at least 0.5 Gy RTX dose before and had shown a plateau in the nucleoid damage effects with more than 5 Gy RTX dose [42].

RTX dose was applied at room temperature using the Xylon Maxi shot (serial number 1210001434) at 200 KV, 10 mA at a rate of 1 Gy/1:08 min. Irradiated cells were further used for Western blot analysis or real time PCR as described above 24 or 48 h after RTX. Radiosensitivity was assessed using the CFA.

### 4.9. Tyrosine Kinase Inhibitor (TKI) R428 Treatment in Cell Culture

R428 (Bemcentinib, BGB324, cat#S2841) was purchased from Selleckchem (Houston, TX, USA) and was dissolved in dimethyl sulfoxide (DMSO, Sigma-Aldrich, St. Louis, MO, USA). Stock solutions of 1 mM were prepared according to the manufacturer’s instructions, and R428 was diluted in Dulbecco’s modified Eagle medium (DMEM) to final concentrations of 0.5 µM, 1 µM, and 2 µM. Human glioma cells were treated with a single dose of R428. 

### 4.10. Statistics

Each group consisted of at least triplicates. Statistical analysis was performed using GraphPad Prism, Version 7.0c. Statistical significance was measured using Student’s *t*-Test and one-way ANOVA combined with Bonferroni’s multiple comparison test. Significance level was set at alpha = 0.05 (95% confidence intervals). The level of significance was set at *p* < 0.05. Data were expressed as mean +/− standard deviation. Quantitative and qualitative statistical analyses were performed using GraphPad Prism 5 (GraphPad Software, La Jolla, CA, USA).

## Figures and Tables

**Figure 1 ijms-23-00982-f001:**
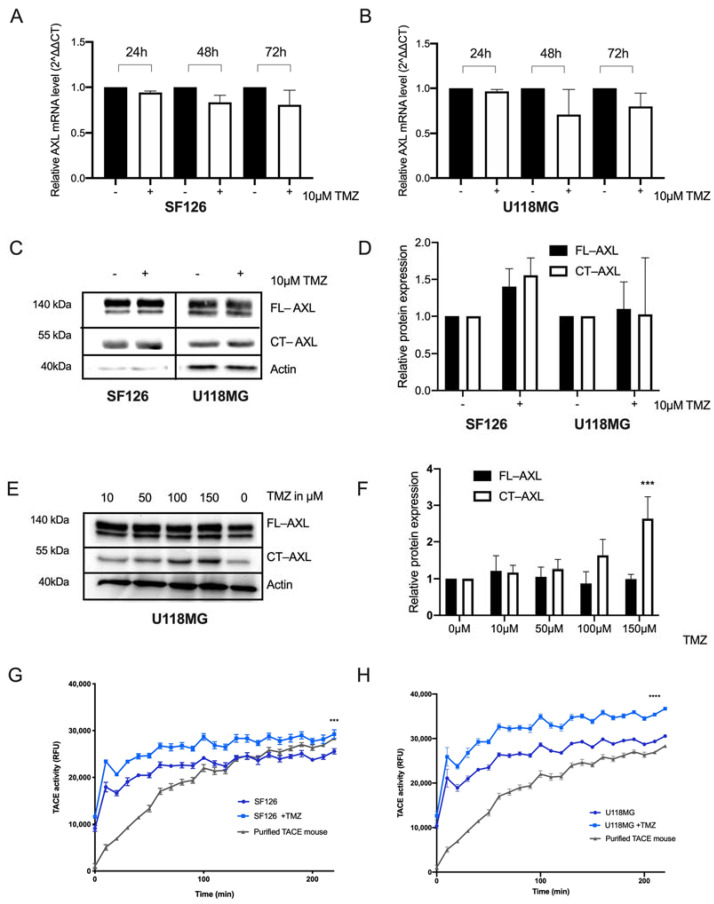
RTK-AXL undergoes post-translational modifications following temozolomide. (**A**,**B**) Fold gene expression changes in SF126; (**A**) und U118MG; (**B**) cell lines following 10 µm TMZ exposure at 24, 48 and 72 h (*p* = not significant); (**C**) Western blot with lysates of SF126 and U118MG cell lines treated with 10 µM TMZ for 48 h displaying full length RTK-AXL (FL-AXL) at 140 kDa and C-terminal RTK-AXL (CT-AXL) at 55 kDa; (**D**) Quantitative analysis of relative protein expression of FL-AXL and CT-AXL normalized to beta actin (*p* = not significant); (**E**) Effect of increasing doses of Temozolomide (TMZ) and repeated exposures (*n* = 5) on protein expression of FL-AXL and CT-AXL expression in U118MG (Western blot) and quantification of protein expression of FL-AXL and CT-AXL expression in U118MG. Ordinary one-way ANOVA, **** *p* < 0.0001 (**F**). (**G**) Activity of the TACE matrix metalloproteinase ADAM 10/17 following TMZ treatment in SF126 (ordinary one-way ANOVA, *** *p* < 0.005) und U118MG (ordinary one-way ANOVA, **** *p* < 0.0001) (**H**).

**Figure 2 ijms-23-00982-f002:**
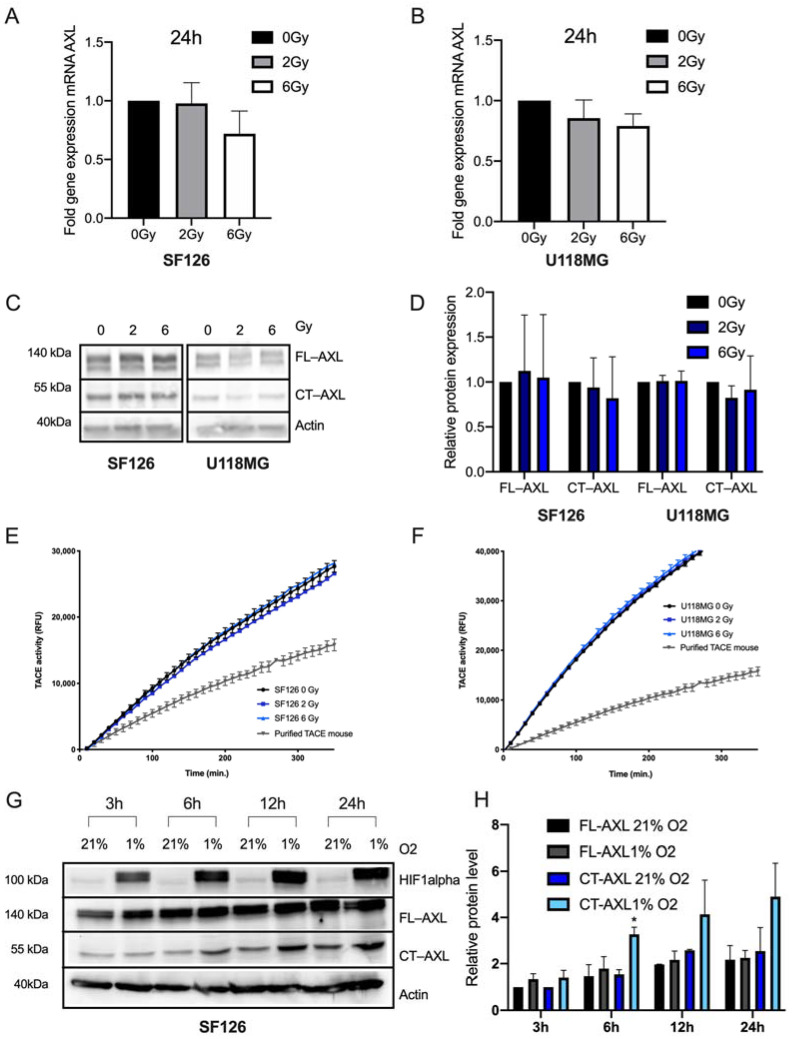
RTK-AXL undergoes post-translational modifications in a radio-resistant microenvironment. (**A**,**B**) Fold gene expression changes in SF126 (**A**) und U118MG (**B**) cell lines following 0, 2 and 6 Gy radiation. Ordinary one-way ANOVA, SF126: *p* = ns, U118MG: *p* = ns. (**C**) Western Blot with lysates of SF126 and U118MG cell lines treated with 0, 2 and 6 Gy radiation displaying full length RTK-AXL (FL-AXL) at 140 kDa and C-terminal RTK-AXL (CT-AXL) at 55 kDa. (**D**) Quantitative Western blot analysis of C (*p* = not significant). (**E**,**F**) Activity of the TACE matrix metalloproteinase ADAM 10/17 following radiation treatment in S126 (**E**) and U118MG (**F**). Ordinary one-way ANOVA, SF126: *p* = ns, U118MG: *p* = ns. (**G**) SF126 cultivated under hypoxia (1% O_2_) for 3, 6, 12 and 24 h displaying HIF1 alpha, FL-AXL and CT-AXL compared to normoxia (21% O_2_). (**H**) Quantification of protein expression of FL-AXL and CT-AXL in U118MG. Student’s *t*-test comparing CT-AXL 21% O_2_ with CT-AXL 1% O_2_, * *p* = 0.023.

**Figure 3 ijms-23-00982-f003:**
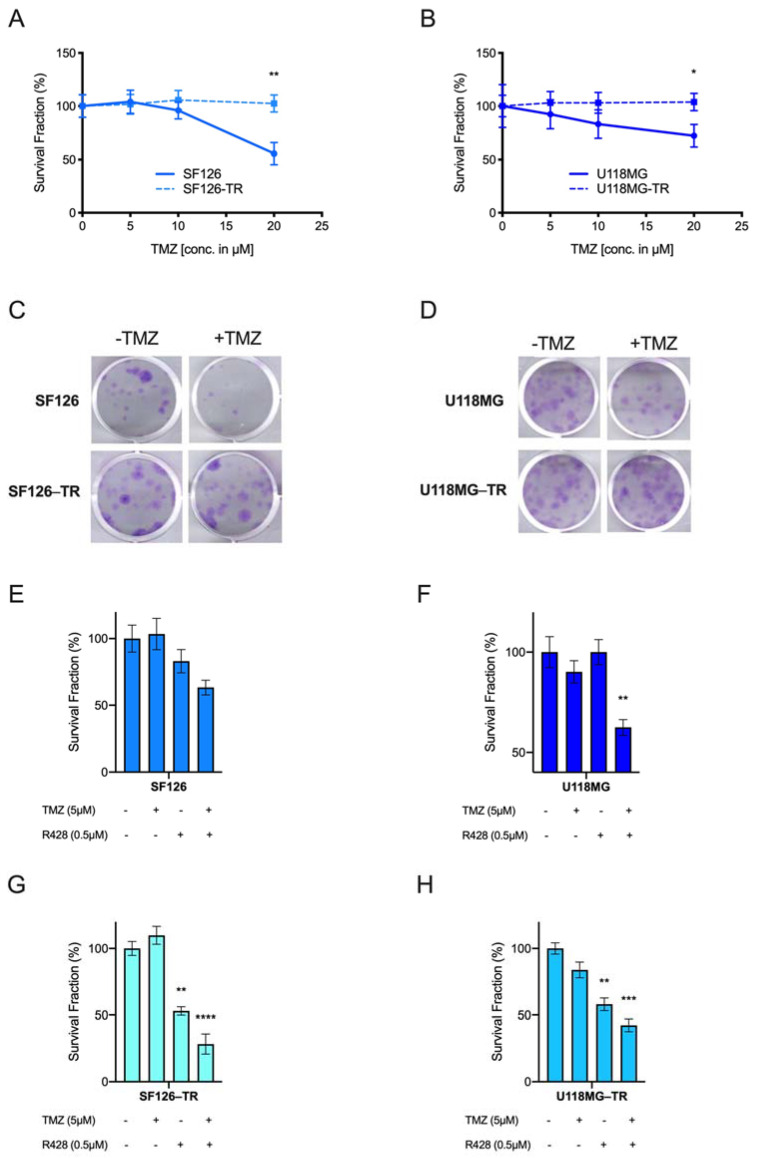
AXL TKI R428 combined with temozolomide (TMZ) increases therapeutic effects: (**A**,**B**) colony formation assay (CFA) of SF126 and SF126-TR (**A**); and U118MG and U118-TR (**B**) cell lines under TMZ exposure. Paired *t*-test SF126 vs. SF126-TR: ** *p* = 0.006; paired *t*-test U118MG vs. U118MG-TR: ** *p* = 0.025; (**C**,**D**) representative images of the CFA of SF126 and SF126 temozolomide-resistant (TR); (**C**) and U118MG and U118MG-TR with and without TMZ treatment (**D**); (**E**) survival fraction of SF126 under TMZ and R428 exposure. Ordinary one-way ANOVA, SF126: *p* = ns; (**F**) survival fraction of U118MG under TMZ and R428 exposure. Ordinary one-way ANOVA U118MG ** *p* = 0.0069; (**G**) survival fraction of SF126-TR under TMZ and R428 exposure. Ordinary one-way ANOVA, SF126-TR: **** *p* < 0.0001. Student’s *t*-test comparing SF126 with SF126 + R428: ** *p* = 0.0015; (**H**) survival fraction of U118MG-TR under TMZ and R428 exposure. Ordinary one-way ANOVA U118MG *** *p* = 0.0002. Student’s *t*-test comparing U118MG with U188MG + R428: ** *p* = 0.0028.

**Figure 4 ijms-23-00982-f004:**
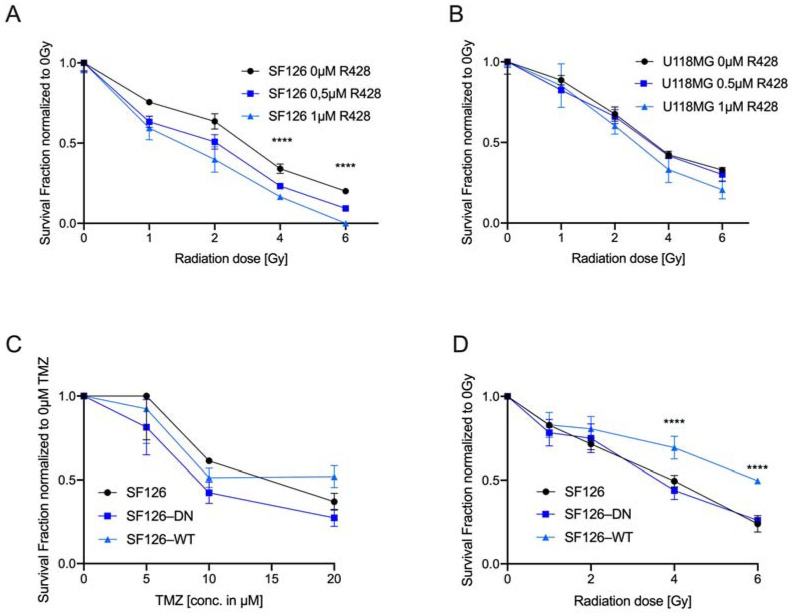
AXL TKI R428 combined with radiation (RTX) increases therapeutic effects and endogenous RTK-AXL overexpression is related to RTX resistance: (**A**) Survival fraction of SF126 under radiation and R428 exposure. Ordinary one-way ANOVA, SF126: **** *p* < 0.0001; (**B**) Survival fraction of U118MG under radiation and R428 exposure. Ordinary one-way ANOVA, U118MG: *p* = ns; (**C**) survival fraction of SF126, SF126-DN and SF126-WT cell lines in colony formation assay under increasing dosage of temozolomide (TMZ). Ordinary one-way ANOVA, *p* = ns; (**D**) survival fraction of SF126, SF126-DN and SF126-WT cell lines in colony formation assay under increasing dosage of radiation (0, 2, 4, and 6 Gy). Ordinary one-way ANOVA, SF126WT at 2 and 6 Gy: **** *p* < 0.0001.

**Table 1 ijms-23-00982-t001:** Primer sequences.

Target	Forward Primer 5′-3′	Reverse Primer 5′-3′
hAXL	GTGGGCAACCCAGGGAATATC	GTACTGTCCCGTGTCGGAAAG
mAXL	ATGGCCGACATTGCCAGTG	CGGTAGTAATCCCCGTTGTAGA
h18s	CATGGCCGTTCTTAGTTGGT	CGCTGAGCCAGTCAGTGTAG
m18s	AACCCGTTGAACCCCATT	CCATCCAATCGGTAGTAGCG

## Data Availability

The presented data in this study are available on request from the corresponding author.

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
