# Peer review of "Regulation of the Receptor Tyrosine Kinase AXL in Response to Therapy and Its Role in Therapy Resistance in Glioblastoma"

_ijms, 2022, doi:10.3390/ijms23020982_

Round 1
Reviewer 1 Report
The manuscript by Scherschinski et al. is an interesting and well-written account on the regulation of the receptor tyrosine kinase AXL in response to therapy and its role in tumor progression in glioblastoma multiforme. There are some details, mainly methodological, that should be addressed in the revised manuscript.
Major problems:
1) The Introduction section seems too short. Perhaps the readers would be better served if more background data, supported by references, were provided about this particular RTK.
2) There is no explanation for radiation exposure, the dose of radiation, etc. The procedure, instrumentation, handling of the cultures during and after irradiation, etc. should be explained in more detail with references to support the choices. Why these doses, some of them large, were used for this experiment? The radiation protocol used here should be compared with other studies using the same or different cell lines (with references).
3) The authors should explain in detail how parallels/independent measurements were set up.
Minor problems:
1) Abbreviations must be spelled out when first used, see line 20 and line 283 for MTT, line 37 for EGFR, NTRK, line 75 for TACE and ADAM, etc., throughout the text.
2) The Abstract should be without headings (see manuscript template).
3) Line 22: “mRNA level” should be used instead of “RNA level”.
4) Line 38: makers of multi-kinase inhibitors should be indicated
5) Lines 58 and 60: in my opinion, the first two paragraphs of the Results section should be the first subsection. As it is now, studies on post-translational modifications of RTK-AXL do not belong to anywhere.
6) The authors did not see any AXL mRNA regulation for TMZ or radiation exposure, and therefore did not show data related to this phenomenon. I believe these observations are important and should be shown to the reader; as the manuscript contains only three Figures (albeit two of them with multiple panels), there is room for such data even if it will end up in the Supplementary Material.
7) Line 67: indication of multiple panels in a Figure should be with commas (Figure 1A, B) instead of the + sign.
8) How was the protein content of the cultures, and those of the samples, determined?
9) Line 235: how did the authors determine the primer sequences? Please provide supporting data and references.
10) The rationale for the use of TKI R428 should be explained with reference(s).
11) Line 64: “short- or long-term exposure (12, 24, 48 and 72h)” seems arbitrary, it should be explained: which one is which?
12) Line 211: how “equal (80%) confluence under normal or starving conditions” were achieved? Would it mean the cells were cultured for different times?
13) Line 214: For the hypoxia experiments, why the cells were cultured exactly for 3, 6, 12, or 24 hours? Please provide references supporting these choices.
Reviewer 2 Report
The authors describe an RTK pathway, AXL, that is modulated in GBM TMZ resistant lines
The preliminary data is interesting yet the data set is still inconclusive
there is no quantification of western blot data
there is no mention of TMZ resistance mediated by AXL knockdown or overexpression
only one mRNA loading control was used in qPCr datasets (18s)
there appears to be only actin as a loading control for protein data
could the authors do sequencing coupled with pathway analysis to determine if AXL pathway is truly modulated due to TMZ
There was minor phenotypic outcomes in TR lines?
could the data be repeated in multiple clones of TR cell lines ?
Please resubmit with adequate controls.
Round 2
Reviewer 2 Report
Thank you for your comments and corrections
to this reviewer it is still not clear how C-AXL is functional
- If cleavage occurs post transcriptionally or post translation?
- Beta actin western controls are quite saturated
overall, without directly modifying C-AXL to alter TMZ response, there it is not clear how much additives value C-AXL is
Round 3
Reviewer 2 Report
The authors have addressed the reviewers concerns